# Immunogenic Cell Death Associated Molecular Patterns and the Dual Role of IL17RA in Interstitial Cystitis/Bladder Pain Syndrome

**DOI:** 10.3390/biom13030421

**Published:** 2023-02-23

**Authors:** Wei Zhang, Xiaodong Liu, Jiawen Wang, Xinhao Wang, Yaoguang Zhang

**Affiliations:** Department of Urology, Beijing Hospital, National Center of Gerontology, Institute of Geriatric Medicine, Chinese Academy of Medical Sciences, Beijing 100730, China

**Keywords:** immunogenic cell death, interstitial cystitis, interleukin 17 receptor, urothelial cells, macrophages

## Abstract

The unclear etiology and pathogenesis of interstitial cystitis/bladder pain syndrome (IC/BPS) are responsible for the lack of effective treatment and the poor patient prognosis. Various studies show that chronic inflammation and immune responses are important factors contributing to the pathogenesis of IC/BPS. The process of immunogenic cell death (ICD) involves both the immune response and inflammatory process, and the involvement of ICD in IC/BPS pathogenesis has not been explored. Two IC/BPS transcriptome datasets collected from the Gene Expression Omnibus (GEO) database were used to identify distinct ICD-associated molecular patterns (IAMPs). IAMPs and IC/BPS subtypes were found to be related. The inflammatory immune microenvironments (IIME) in different IAMPs were studied. The potential mechanism by which the interleukin 17 receptor A (IL17RA) influences IC/BPS was examined using in vitro assays. The expression of ICD-related genes (IRGs) was upregulated in IC/BPS bladders, compared with normal bladders. Disease prediction models, based on differentially expressed IRGs, could accurately predict IC/BPS. The IC/BPS patients had two distinct IAMPs, each with its own subtype and clinical features and association with remodeling IIME. IL17RA, a well-established IC/BPS bladder biomarker, mediates both the inflammatory insult and the protective responses. In summary, the current study identified different IAMPs in IC/BPS, which may be involved in the pathogenesis of IC/BPS by remodeling the IIME. The chronic inflammatory process in IC/BPS may be prolonged by IL17RA, which could mediate both pro- and anti-inflammatory responses. The IL17RA-associated pathway may play a significant role in the development of IC/BPS and can be used as a therapeutic target.

## 1. Introduction

Interstitial cystitis/bladder pain syndrome (IC/BPS) is a clinical syndrome that primarily affects the urinary bladder and causes debilitating pain and lower urinary tract symptoms. Based on the endoscopic findings, there are currently two main subtypes of IC/BPS: IC/BPS patients with a Hunner’s lesion (HIC) and IC/BPS patients without a Hunner’s lesion (NHIC). The latter subtype can be further divided into patients without Hunner’s lesions, but with glomerulations, and patients without Hunner’s lesions or glomerulations [1]. IC/BPS adversely affects the life quality of patients [2]. However, due to unclear IC/BPS pathogenesis, the condition is difficult to diagnose and lacks effective treatment strategies for now, which poses a major challenge for the urologist.

Recent studies, including our single-cell sequencing analysis, found that extensive remodeling of the inflammatory immune microenvironment (IIME) played an important role in the pathogenesis of IC/BPS [3,4], although the exact mechanism of IIME remodeling involved is yet unclear. Immunogenic cell death (ICD), defined as a novel form of cell death, closely related to the adaptive immune response triggered by certain types of regulated cell death (RCD) under specific conditions [5,6], contributed to the pathogenesis of various IIME remodeling-associated diseases, such as cancer and rheumatic disease [5,7,8]. However, whether ICD was involved in the IIME remodeling of IC/BPS has never been reported yet, although various forms of RCD (such as apoptosis and pyroptosis) participated in the development of IC/BPS [9,10]. In addition, interleukin 17 (IL17) and interleukin 17 receptor A (IL17RA) are currently considered to be the important mediators of ICD [11], among which, IL17RA mediates the biological function of most IL17 family members [12]. The IL17 family has also been reported to play a key role in many IIME remodeling-associated diseases, such as psoriasis, yet the association between IL17RA and IC/BPS remains to be further clarified.

It was, thus, hypothesized that ICD may participate in the IIME remodeling of IC/BPS patients. In order to explore the potential association between ICD and IC/BPS and the underlying mechanisms, a series of bioinformatical algorithms and in vitro experiments were performed to identify the most valuable ICD-related genes (IRGs) in the diagnosis and pathogenesis of IC/BPS.

## 2. Materials and Methods

### 2.1. Data Collection

A total of three datasets, GSE11783, GSE57560, and GSE621 (https://www.ncbi.nlm.nih.gov/geo/query/acc.cgi?acc=GSE11783,GSE57560,GSE621, accessed on 1 August 2022), containing transcriptome data and clinical information on bladder tissues from IC/BPS patients and normal individuals, were retrieved from the Gene Expression Omnibus (GEO) database. The dataset GSE621 was excluded from the study, as it contained too few genes (*N* = 3353). The datasets GSE11783 and GSE57560 were merged using the ‘SVA’R package to eliminate batch effects. All data used in the study were obtained from publicly available databases, such as GEO; therefore, ethical approval and informed consent were not required. The ICD-related genes (IRGs) expression profiles, which are summarized by Garg et al. [13], were extracted from the merged datasets.

### 2.2. Identification of the IRGs Signature

A total of 34 IRGs were included in the analysis. Differential expressions of IRG between normal and IC/BPS bladder tissues were obtained using the Wilcoxon test function [14]. To correct the *p* value, Benjamini and Hochberg (BH) adjustment was used. Adjust *p* < 0.05 was set as the threshold for selection. The differentially expressed IRGs were mapped on human chromosomes and visualized using the ‘RCircos’ R package. The correlation between IRGs was analyzed using the ‘Corrplot’ package in R software [15].

### 2.3. Construction of the Diagnostic Prediction Model

Random forest (RF) [16] and support vector machine (SVM) [17] models were used to construct diagnostic prediction models for IC/BPS. The ‘randomForest’ R package was used to build the RF model, and the ‘kernlab: Kernel-Based Machine Learning Lab’ R package was used to build the SVM model, based on the differentially expressed IRGs. “Reverse cumulative distribution of residual,” “Boxplots of residual,” and receiver operating characteristic (ROC) curves were plotted to determine the best model.

### 2.4. Construction of Nomogram Models

A nomogram model was constructed based on the screened IRGs to predict the prevalence of IC/BPS, using the ‘rms’ package in R software [18]. The consistency of the predicted values against reality was evaluated using the calibration curve. Decision curve analysis (DCA) was performed, and clinical impact curves were plotted to evaluate whether the decisions made based on the model were useful for the patients [19]. The ROC curves were plotted using the ‘pROC’ R package to evaluate the diagnostic efficacy of each element of the model [20].

### 2.5. Construction of ICD-Associated Molecular Patterns

Consensus clustering was performed using the ‘ConsensusClusterPlus’ in R software to identify the different ICD-associated molecular patterns (IAMPs) [21]. Kyoto Encyclopedia of Genes and Genomes (KEGG) pathways and Gene Ontology (GO) enrichment analyses were performed between different IAMPs using the ‘Gene Set Variation Analysis (GSVA)’ R package [22]. An adjusted *p* < 0.05 was considered as a criterion for screening. Single sample gene set enrichment analysis (ssGSEA) [23] was used to evaluate the difference in immune cell infiltration, immune function, HLA gene expression, and the levels of different RCD of distinct IAMPs.

### 2.6. Identification of Differentially Expressed Genes between Different IAMPs and the Functional Enrichment Analysis

The ‘limma’ R package was used to screen for differentially expressed genes (DEGs) between the different IAMPs [24]. An adjusted *p* < 0.05, |logFC| > 2 was used as the screening criterion to select DEGs. GO enrichment analysis and KEGG pathway enrichment analysis of DEGs were performed using the ‘clusterProfiler’ package in R software to explore the differential mechanisms associated with distinct IAMPs [25].

### 2.7. Conjoint Analysis of Weighted Gene Correlation Network Analysis and DEGs between Distinct IAMPs

Weighted gene correlation network analysis (WGCNA) analysis was performed on different IAMPs using the ‘WGCNA’ software of R package to explore the modules (ICD module genes) related to different IAMPs [26]. GO enrichment and KEGG pathway enrichment analysis were used to understand the different IAMP mechanisms, using the intersection of DEGs and ICD module genes. The analysis was performed using the ‘clusterProfiler’ R package in R software. The protein–protein interaction (PPI) was analyzed with the Search Tool for the Retrieval of Interacting Genes/Proteins (STRING) database (https://cn.string-db.org/, accessed on 1 August 2022), and the cytoHubba plug-in of Cytoscape software (version 3.8.2) was used to explore hub genes.

### 2.8. Quantification of IAMPs and Assessment of Their Correlation with IC/BPS Subtypes

The principal component analysis (PCA) algorithm was used to calculate the ICD score of each sample to quantify the IAMPs. The prcomp function of ‘Stats’ R package was used to compute PCA with the default settings [27]. The ‘plyr’ R package was used to evaluate the correlation between ICD scores and different IAMPs [28]. Based on DEGs, consensus clustering was used to identify IAMPs-related gene clusters. Sankey diagrams of different IAMPs, gene clusters, and IC/BPS subtypes were plotted using the ‘ggalluvial’ R package [29]. The ‘ESTIMATE’ R package was used to analyze the differences in the immune microenvironment of different IAMPs, gene clusters, bladder volume, and IC/BPS subtypes [30]. In short, we used the ESTIMATE algorithm to evaluate the different components of bladder tissue in patients with IC/BPS. The stromal score represents the matrix component of bladder tissue, the immune score represents the infiltration of immune cells in bladder tissue, and the ESTIMATE score represents the purity of bladder tissue.

### 2.9. Cell Culture and Transfection

The normal human urothelial cell line (SV-HUC-1) and mouse macrophage cell line (RAW264.7) were purchased from Procell Life Science & Technology Co., Ltd.(Wuhan, Hubei, China). SV-HUC-1 and RAW264.7 cells were cultured in Kaighn’s modification of Ham’s F-12 (F-12K) medium (Gibco, Carlsbad, CA, USA) and Dulbecco’s modified Eagle medium (DMEM) medium (Gibco, Carlsbad, CA, USA), respectively. The F-12K and DMEM medium were supplemented with 10% fetal bovine serum (FBS) and 1% penicillin-streptomycin solution (both purchased from Gibco, Carlsbad, CA, USA) and cultured at 37 °C with 5% CO2 and 95% relative humidity. The cells were passaged at 60–70% confluency. pcDNA3.1-IL17RA was used to overexpress IL17RA, which was constructed using the pcDNA3.1/His B plasmid, and the blank plasmid (pcDNA3.1-NC) was used as a control. Before transfection, the cells were cultured in an antibiotics-free medium, and the transfection was performed using Lipofectamine 2000 (Invitrogen, Carlsbad, CA, USA) per the manufacturer’s instructions. Lipopolysaccharide (LPS) was used to induce inflammatory damage. SV-HUC-1 cells were divided into the IL17RA-NC group, IL17RA-OE group, IL17RA-NC-LPS group, and IL17RA-OE-LPS group. The IL17RA-NC group and IL17RA-NC-LPS group were transfected with the control plasmid pcDNA3.1-NC, and the IL17RA-OE group and IL17RA-OE-LPS group were transfected with a plasmid containing the target gene pcDNA3.1-IL17RA.

### 2.10. Total RNA Extraction and Quantitative Real-Time Polymerase Chain Reaction

Total RNA was extracted using TRIzol reagent (Invitrogen, Carlsbad, CA, USA) per the manufacturer’s instructions. RNA was reverse transcribed using HiScript^®^ II Q RT SuperMix (Vazyme Biotech, Nanjing, Jiangsu, China) to synthesize complementary DNA (cDNA). Quantitative real-time polymerase chain reaction (qRT-PCR) was performed using 2 × RealStar Fast SYBR qPCR Mix (GenStar, Beijing, China) on a 7500 real-time PCR system (Applied Biosystems, Foster City, CA, USA). ACTB (β-Actin) was used as a housekeeping gene. The primer sequences are listed in Appendix A.

### 2.11. Cell Proliferation Assays

Cell proliferation ability was performed using xCELLigence Technology (ACEA Bioscience, San Diego, CA, USA). A total of 4000 SV-HUC-1 cells/well were seeded in E-16-well plates (ACEA Bioscience, San Diego, CA, USA), and transfection was performed post 36 h of culturing the cells. IL17RA-NC-LPS group and IL17RA-OE-LPS group were treated with LPS (5 μg/mL) 36 h after transfection. Cell growth was monitored for another 24 h. Microelectrodes were placed at the bottom of the E-16 plate to detect changes in impedance proportional to the number of adherent cells. The impedance values for each well were automatically monitored by the xCELLigence system and expressed as cell index values.

### 2.12. Wound Healing Assays

SV-HUC-1 cells were seeded in 6-well plates. The cells were transfected and cultured for 24–48 h. Once the cells achieved 100% confluency, a wound was created using a 200 μL pipette tip. The culture medium was removed. The cells were washed with phosphate-buffered saline (PBS), and 2 mL of F-12K medium without FBS was added to each well (an additional 5 μg/mL of LPS was added to the IL17RA-NC-LPS group, and IL17RA-OE-LPS group). Images were captured immediately after media change was noted as t = 0, and the images were taken every 12 h for the next 36 h using a microscope (OLYMPUS, Tokyo, Japan). The wound area was measured using ImageJ software, and the wound closure was measured at T0, T12, T24, and T36.

### 2.13. Interactions between Urothelial Cells and Macrophages

The interaction between urothelial cells and macrophages was detected using the conditioned media co-culture system. Briefly, 36 h post-transfection, SV-HUC-1 cells were treated with vehicle (IL17RA-NC group and IL17RA-OE group) or LPS (5 μg/Ml, IL17RA-NC-LPS group, and IL17RA-OE-LPS group). After 24 h of treatment, the cells were washed with PBS, and a fresh medium was added. The culture medium from the four treatment groups was collected after 12 h. RAW264.7 cells were treated with the conditioned medium of the four groups for 24 h. The expression of genes associated with macrophage polarization and phagocytosis was determined by qRT-PCR.

### 2.14. Chronic LPS Stimulation’s Influence on Urothelial Cells

As previously described: after 36 h of transfection, four groups of SV-HUC-1 cells were treated with vehicle (IL17RA-NC group and IL17RA-OE group) or LPS (5 μg/mL, IL17RA-NC-LPS group, and IL17RA-OE-LPS group), respectively. The medium was changed to fresh medium without LPS after 24 h, and the incubation was continued for another 12 h. The purpose of the above treatments was to mimic the microenvironment in which the cells recovered from LPS injury. Finally, the changes of pro-inflammatory-, anti-inflammatory-, and barrier function-related genes in urothelial cells of different groups were examined.

### 2.15. Statistical Analysis

All bioinformatics analyses were performed using R software 4.1.3 (Vienna, Austria). Statistical analysis was performed using GraphPad Prism 8.0 (GraphPad Software, San Diego, CA, USA). In vitro data was represented as mean ± SEM (standard error of the mean). One-way ANOVA was used for multiple comparisons. *p* < 0.05 was considered statistically significant.

## 3. Results

### 3.1. IRGs Expression Signature in IC/BPS Patients

The two datasets (GSE11783 and GSE57560) retrieved from the GEO data were merged. The dataset contained information on the expression profile of the bladder tissues obtained from nine normal individuals and 23 IC/BPS patients. Differential expression was observed in 19 IRGs between IC/BPS patients and normal individuals (Figure 1a). Except for HSP90AA1 and NT5E, most IRGs were overexpressed in IC/BPS patients, compared to normal individuals (Figure 1b), and the difference in expression profile within IC/BPS patients suggests the possibility of different IAMPs. Figure 1c shows the chromosomal location of the IRGs. Among the IRGs expressed in the bladder tissues of IC/BPS patients, only ATG5, EIF2AK, IFNB1, HSP90AA1, and NT5E were negatively correlated with most of the other IRGs (Figure 1d).

### 3.2. Construction of RF and SVM Model

RF and SVM models were constructed on the basis of the differential expression of IRGs between IC/BPS patients and normal individuals, which could predict the occurrence of IC/BPS. The ‘reverse cumulative distribution of residual’ (Figure 2a) and ‘boxplots of residual’ (Figure 2b) showed that the RF model has smaller residuals, indicating that RF was a better prediction model than SVM. Hence, the RF model was used to predict the occurrence of IC/BPS. The IRGs predicted by the RF model were ranked according to their importance (Figure 2c). The top five IRGs (IL17RA, IL1B, NT5E, IL17A, and CASP1) were selected as model genes. Finally, the ROC curve analysis confirmed that the RF model had a better prediction accuracy than the SVM model (Figure 2d).

### 3.3. Construction of a Nomogram Model

A nomogram model was constructed to predict the prevalence of IC/BPS patients based on the five IRGs model genes (Figure 3a). The calibration curve showed the predictive performance of the nomogram model, demonstrating that the prediction of the model was in good alignment with the actual situation (Figure 3b). The red line in the DCA curve remained above the gray and black lines, indicating that decisions made based on the constructed nomogram model could benefit the IC/BPS patients (Figure 3c). The clinical impact curve showed that the nomogram model had a good predictive ability (Figure 3d). The ROC curves of the model genes indicated that all the five IRGs had good predictive ability, among which, IL17RA had the best predictive ability among all the five IRGs, with an area under curve (AUC) of 0.913 (Figure 3e).

### 3.4. Identification of Molecular Subtypes Based on the Differentially Expressed IRGs

Consensus clustering was used to identify two ICD molecular subtypes, which were defined as IAMPs (ICD cluster A and ICD cluster B), based on the pool of 19 differentially expressed IRGs between IC/BPS and normal individuals (Figure 4a). ICD cluster A contained 14 cases, and ICD cluster B had nine cases. PCA analysis demonstrated good discrimination of the two IAMPs (Figure 4b). Figure 4c,d reveal a significant difference in the expression of 14 IRGs between ICD cluster A and B. All differentially expressed IRGs, besides HSP90AA1, showed higher expression levels in the ICD cluster A than in the ICD cluster B.

### 3.5. Identification of DEGs between Distinct IMAPs and Functional Enrichment Analysis

To explore the mechanisms of different IAMPs, GO enrichment and KEGG pathway enrichment analyses were performed on the DEGs in ICD cluster A and B. The GO analysis revealed that these DEGs were associated with the activation, adhesion, and proliferation processes of T cells and other inflammatory or immune cells (Figure 5a,b). Similar to GO analysis, the enriched pathways from KEGG analysis were related to inflammation and immune functions, with the cytokine–cytokine receptor interaction being the most significant pathway (Figure 5c). Since most enriched pathways of the DEGs were significantly correlated with immune and inflammation, immune characteristics were then compared between clusters using ssGSEA. Figure 5d showed that most immune cells were more activated in cluster A, and the further comparison of immune responses and expression of HLA genes drew similar conclusions (Appendix A), suggesting that ICD cluster A may be associated with more activated IIME. Additionally, a positive correlation was observed between most IRGs and various infiltrated immune cells. Only HSP90AA1, the expression of whom was lower in cluster A, was negatively correlated with the immune cells, further indicating that cluster A had a more highly activated immune status. Further, immature dendritic cells and CD56 bright NK cells negatively correlated with the large fraction of IRGs (Figure 5e). Finally, the features of the five common RCD modalities in different ICD clusters were analyzed through ssGSEA, and the results showed that the apoptosis, autophagy, pyroptosis, and necroptosis were more activated in ICD cluster A than in ICD cluster B, while ferroptosis showed no significant difference between clusters (Figure 5f).

### 3.6. Conjoint Analysis of WGCNA and DEGs of Different IAMPs

To further explore the function difference between the distinct IAMPs, a conjoint analysis of WGCNA and DEGs was performed. During WGCNA analysis, the turquoise module was identified as the module most significantly and positively associated with ICD cluster A, which had a more activated IIME (Figure 6a). Then, overlapping of the genes from the turquoise module with DEGs between the two IAMPs was conducted, and 93 intersection genes were identified (Figure 6b). Further, the GO and KEGG analyses of these intersection genes showed that the inflammatory and immune pathways were most highly enriched and the cytokine–cytokine receptor interaction pathway was still the most important pathway, consistent with the former analyses (Figure 6c–e). Finally, PPI analysis was performed on the intersection genes, and antigens expressed by various inflammatory immune cells, such as CD19, CCR7, CD27, and CTLA-4, were among the top ten hub genes (Figure 6f,g). The conjoint analysis further confirmed that the IIME status differed between distinct IAMPs.

### 3.7. Identification of Different IAMPs Genomic Characteristics

The IC/BPS patients were classified into two distinct genomic subtypes, based on ICD-associated DEGs, using consensus clustering (Figure 7a,b). Figure 7c,d showed that the expression of IRGs and immune cell infiltration in different gene clusters were largely consistent with IAMPs. These results further validated the accuracy of the IAMPs classification.

### 3.8. ICD Score and Correlation with HIC/NHIC Subtypes

PCA was used to assess the ICD scores of each sample to quantify the IAMPs (ICD score = ∑(principal element 1e + principal element 2e), where e is defined as the expression of IRGs) 31. Further, the ICD scores were compared between the two different ICD clusters or gene clusters. The results showed that ICD cluster A or ICD gene cluster A had significantly higher ICD scores than ICD cluster B or ICD gene cluster B (Figure 8a,b). Meanwhile, the HIC subtype had a significantly higher ICD score, compared to the NHIC subtype (Figure 8c,d). In addition, the Sankey diagram revealed that the NHIC subtype comprised the great majority of IC/BPS patients in ICD cluster B or ICD gene cluster B, while almost all HIC patients belonged to ICD cluster A or ICD gene cluster A. (Figure 8e). The consistency between these molecular subtypes and clinical subtypes of IC/BPS prominently suggested the importance of ICD in the pathogenesis of IC/BPS.

### 3.9. Correlation of Immune Infiltration Scores with Distinct IAMPs and Clinical Features

ESTIMATE score analysis was performed to explore the differences in the immune microenvironment of different IAMPs and clinical features. Figure 9a–f showed high stromal cell and immune cell infiltration scores in the ICD cluster A and ICD gene cluster A, implying that the ICD cluster A and ICD gene cluster A may have more stromal cell and immune cell infiltration. Further, IC/BPS patients with NHIC subtype or normal bladder capacity had low stromal cell and immune cell infiltration scores, as seen in Figure 9g–l, indicating less stomal and immune cell infiltration.

### 3.10. IL17RA Expression in IC/BPS and Correlation with IIME

IL17RA showed the highest predictive value, with the largest AUC among the significant IRG predictors; hence, further comprehensive analysis of IL17RA was performed to explore its potential role in the IIME of IC/BPS patients. Compared to normal individuals, higher IL17RA expression was observed in the bladder tissues of IC/BPS patients (Figure 1a). Further, higher IL17RA expression was observed in ICD cluster A, compared to ICD cluster B (Figure 4c). Additionally, a positive correlation was observed between IL17RA and the expression of most IRGs (Figure 1d). Of note, IL17RA showed the most significant correlation with TNF (R = 0.85), a major diagnostic marker and treatment target for IC/BPS [31,32,33]. The ssGSEA analysis indicated that IL17RA was positively correlated with the large fraction of infiltrated immune cells, so was TNF (Figure 5e). Our earlier research revealed that the interaction between macrophages and endothelial cells might be a prominent characteristic of IC/BPS [4]. Of note, a significant positive correlation was observed between the IL17RA expression and macrophage infiltration in IC/BPS patients (Figure 10a). Additionally, high concordance was observed between IL17RA expression and TNF, TLR4, NLRP3, IL1B, and IL10, regarding correlation with immune characteristics (Figure 5e and Figure 10b,c).

### 3.11. Effects of Overexpression of IL17RA in Urothelial Cells on LPS-Induced Inflammatory Proliferation and Migration Capacity

The qRT-PCR results showed that the IL17RA was successfully overexpressed in SV-HUC-1 cells as compared to control cells after transfection, while LPS treatment significantly increased IL17RA expression (Figure 11a), compared to untreated cells. Overexpression of IL17RA significantly inhibited the proliferative capacity of SV-HUC-1 cells after an injury induced by LPS (Figure 11b). However, wound healing assays revealed that IL17RA did not significantly affect the migratory ability of urothelial cells after LPS injury (Figure 11c). In general, the overexpression of IL17RA impaired urothelial cells’ ability to proliferate, but not their ability to migrate.

### 3.12. The Dual Role of IL17RA in Regulating the Damage-Repair Function of Urothelial Cells

Consistent with the results obtained from the GEO datasets, IL17RA overexpression in urothelial cells (SV-HUC-1) significantly upregulated the TLR4-NLRP3-IL1B inflammatory pathway, especially after LPS stimulation (Figure 12a). Additionally, IL6 expression was increased after IL17RA overexpression, while no significant difference was observed in the expression of NFKB1, TNF, and IL18, with or without LPS application. In addition to the above-mentioned proinflammatory factors, IL17RA overexpression also significantly increased the expression of the anti-inflammatory factor IL10 (Figure 12a). Of note, the upregulation of the non-canonical pyroptosis pathway factor CASP4, after LPS stimulation, could be suppressed with the over-expression of IL17RA, although its expression showed no change after the transfection of IL17RA under normal circumstances (Figure 12a). IL17RA overexpression did not alter the expression of the genes associated with barrier function in urothelial cells during the early stages of inflammatory injury, i.e., 24 h post-treatment with LPS (Figure 12b). Further, the treatment of macrophage cells (RAW264.7) with conditioned media derived from cells overexpressing IL17RA and control cells did not alter the expression of genes associated with phagocytosis and polarization of macrophages. Interestingly, macrophage cells (RAW264.7) treated with conditioned media derived from cells overexpressing IL17RA stimulated with LPS showed a significant increase in the expression of genes associated with M2 polarization and phagocytic function, compared with the control cells when stimulated with LPS (Figure 12c). In addition to IL10, an increase in GLI1 and BMP4 expression was observed in IL17RA overexpressing cells treated with LPS, compared to control cells treated with LPS (Figure 12d), which may lead to glomerulations in the bladder or macrophage polarization.

In sum, IL17RA may play a dual role in IC/BPS pathogenesis. On the one hand, IL17RA upregulates TLR4-NLRP3-IL1B pathways, thereby triggering pro-inflammatory responses. On the other hand, IL17RA increases IL10 expression and induces M2 polarization of the macrophage to counteract the inflammation caused by chronic inflammation and promote tissue repair (Figure 12e).

## 4. Discussion

IC/BPS is a heterogeneous clinical syndrome of unclear etiology and pathogenesis. Chronic inflammation and aberrant immune responses have long been associated with the development and progression of IC/BPS [3,31]. ICD was known as a novel form of cell death, closely related to the adaptive immune response triggered by certain types of RCD under specific conditions, such as stress. A previous report has demonstrated the involvement of pyroptosis, a typical form of RCD, in the pathogenesis of IC/BPS [9]. However, the involvement of ICD in the development, progression, and pathogenesis of IC/BPS has not been investigated. To the best of our knowledge, our study established the correlation between IC/BPS and ICD for the first time.

Although IC/BPS is universally known to severely impair the psychosocial functioning and life quality of patients, the diagnosis and treatment of the disease are still a serious clinical challenge, due to unclear pathogenesis [34]. The current study established a significant ICD model after a series of bioinformatical analyses to best distinguish IC/BPS, with IL17RA being the best predictor, which would facilitate clinical diagnosis to some extent (Figure 1, Figure 2 and Figure 3). Additionally, the predictive value of ICD for IC/BPS also indicated the potential role of ICD in the pathogenesis of IC/BPS. Unfortunately, the model could not be validated in an independent dataset, due to the lack of available datasets on IC/BPS. In order to explore the underlying mechanisms by which ICD participated in the pathogenesis of IC/BPS, consensus clustering of IC/BPS patients was performed based on the 19 differentially expressed IRGs, and two different IAMPs were identified: ICD cluster A and ICD cluster B. A high expression of most IRGs was observed in ICD cluster A, except for HSP90AA1, and those upregulated genes were positively correlated to immune features, suggesting that ICD cluster A had the potential to induce ICD and, thus, had a more activated IIME. Interestingly, HSP90AA1 showed lower expression both in the bladder tissue of IC/BPS patients, compared with normal controls and in cluster A, compared with cluster B. Meanwhile, the correlation between IRGs and immune characteristics indicated that HSP90AA1 was negatively correlated with the activation of immune cells, activation of immune responses and HLA expressions in IC/BPS patients. Therefore, the lower expression of HSP90AA1 still indicated the more activated immune status in cluster A. HSP90AA1 belongs to the family of heat shock proteins (HSPs), which play crucial roles in innate and adaptive immunity and are closely linked to the pathophysiology of numerous immune-mediated illnesses [35]. It has been discovered that HSP90AA1 expression was elevated in the peripheral blood mononuclear cells of a number of autoimmune diseases, including ankylosing spondylitis, systemic lupus erythematosus, and fibromyalgia syndrome [36,37,38,39,40], which was seemingly inconsistent with our results that HSP90AA1 was downregulated in IC/BPS. However, mounting proof indicated that various factors could influence the regulatory effect of HSPs, such as the physio-pathological environment, the concentration of HSPs, and action time, among other factors [36,37,38,39,40,41,42]. In addition, the tissue difference may also account for this discrepancy. So far, there are very few studies on the relationship between HSP90AA1 and IC/BPS, and further in-depth research in this area is intriguing.

GO and KEGG enrichment analysis indicated that the DEGs between distinct ICD clusters were associated with the activation, adhesion, and proliferation of various inflammatory immune cells. Consistent results were obtained via the conjoint analysis of WGCNA and DEGs. Among the top ten hub genes derived from PPI analysis, markers such as CD19, CD79, and IRF4 were mainly expressed on B cells, and CCR7, CD27, CTLA4, CD38, and SELL markers were mainly expressed on T cells. CXCL9 and CXCL10 were primarily responsible for recruiting lymphocytes, monocytes, and macrophages. The ESTIMATE score revealed abundant stromal cell and immune cell infiltration in the ICD cluster A, and ssGSEA analysis showed that various infiltrated immune cells were highly activated in ICD cluster A. These results suggested the activation of IIME might be involved in mediating the different IAMPs. Of note, immature dendritic cells and CD56 bright NK cells were negatively correlated with most IRGs highly expressed in cluster A. It has been reported that immature dendritic cells mediated immune tolerance by regulating the differentiation of T cells [43]. Similarly, CD56 bright NK cells were involved in the immunoregulatory process, such as establishing immune tolerance during early pregnancy [44]. The negative correlation between these two immune-tolerant cells and IRGs highly expressed in cluster A supported the former results that cluster A was more immune-activated. The induction of ICD in tumor cells has been widely confirmed to be able to break the body’s immune tolerance to tumor cells [45]. The intrinsic relationship between ICD and immune tolerance in non-neoplastic diseases has not yet been investigated. As a matter of fact, a growing amount of studies have shown that damage to normal cells can also trigger immunological responses and that a breach in the immune system’s tolerance to normal cells can result in a range of autoimmune disorders [46]. The finding that cells with immune tolerance properties (e.g., immature dendritic cells and CD56 bright NK cells) are negatively associated with IRGs suggested new possibilities: ICD occurring in IC/BPS is accompanied by suppression of immune tolerance, which promoted the autoimmune process in IC/BPS. Additionally, the consistency of highly enriched immune features and more-activated immune RCD modalities (including apoptosis, autophagy, pyroptosis, and necroptosis) in ICD cluster A further supported our hypothesis that ICD- and RCD-related mechanisms could be involved in the pathogenesis of IC/BPS.

Given the clinical relevance and implication of ICD clusters, Sankey diagrams were constructed. The result turned out that IC/BPS patients in the ICD cluster B or ICD gene cluster B subgroups belonged to the NHIC subtype, whereas HIC subtype IC/BPS patients could be grouped in the ICD cluster A or gene cluster A. Additionally, ESTIMATE analysis found that IC/BPS patients with HIC subtype or decreased bladder capacity had significantly higher ESTIMATE scores, which indicated an active IIME. In summary, a more active IIME was associated with ICD cluster A, HIC, and lower bladder capacity. The consistency between IAMPs and IC/BPS clinical subtypes may provide a novel method of personalized intervention for IC/BPS patients, especially for the HIC subtype. It was also noteworthy that the PCA analysis showed that ICD cluster A had greater intragroup variability. Combined with the association between ICD clueter A and HIC subtypes, we have reason to speculate that different subgroups may still exist within HIC patients. It is well-known that inducing ICD in tumors can increase the effectiveness of anti-tumor therapies, whereas inhibiting ICD in non-neoplastic disorders is important for preventing self-injury. Since IIME is more active in IC/BPS patients with HIC or decreased bladder capacity, immunosuppressants may be more suitable for these two subgroups. Consistently, previous research showed that cyclosporine A, the most widely used oral immunosuppressant for IC/BPS, was more effective in HIC patients than NHIC patients [47].

To further understand the mechanism associated with IIME alterations, we explored the involvement of IL17RA, an established IC/BPS marker in regulating bladder function using in vitro model system. Previous studies have shown that IL-17 family is required to maintain homeostasis during injury, physiological stress, and infection [48,49]. Mounting evidence suggests the IL17 family promotes protective immunity against various pathogens and triggers inflammation during infection and autoimmunity [50]. However, the involvement of IL17 signaling in the IC/BPS pathogenesis is still unknown. Our results show an increase in IL17A and IL17RA expression in IC/BPS patients, compared to normal individuals (Figure 1a), and a high expression of IL17RA was observed in ICD cluster A than in ICD cluster B (Figure 4c). This suggests the upregulation of IL17RA could be associated with IC/BPS and is worthy of further exploration. A positive correlation was observed between IL17RA and various IRGs (Figure 1d), exemplifying the intrinsic link between IL17RA and the ICD in IC/BPS. Further, we discovered a positive correlation between IL17RA and the infiltration of immune cells, especially macrophages (Figure 5e and Figure 10a). These results further confirm our previous findings, where the HIC subtype had abundant M2/M2-like macrophages, and M1-like macrophages were found in NHIC and the endothelial cell-macrophage regulatory network in IC/BPS [4]. Furthermore, our study discovered that HIC patients primarily corresponded to ICD cluster A, while IL17RA expression was higher in ICD cluster A, and overexpression of IL17RA in urothelial cells promoted macrophage M2 polarization, validating the association of HIC subtypes with macrophage M2 polarization from a different angle. In addition, TNF has been proposed as a marker and is currently a target for pharmacological intervention in IC/BPS [31,32,33].

The proinflammatory mediator TNF, which is produced by tissue-resident macrophages, was also identified in our earlier research as a critical mediator in the pathological process of IC/BPS [4]. Interestingly, the IL17RA expression pattern and its correlation with immune cell infiltration, immune function, and HLA gene expression pattern are similar to TNF (Figure 5e and Figure 10b,c). TNF and IL17RA have been shown to have synergistic effects in psoriasis, and both are approved by the U.S. Food and Drug Administration as pharmacotherapeutic targets in psoriasis [51]. In light of the above data, it is reasonable to hypothesize that TNF and IL17RA may have also contributed to the remodeling of the IIME in IC/BPS patients, with the potential for combined intervention.

Recently, it has been shown that IC/BPS patients have reduced bladder capacity, increased glomerulations, or Henner’s lesion, which often have significant infiltration of inflammatory cells, denudation of urinary epithelium, and granulation tissue [52]. Therefore, to understand the mechanism associated with the above pathogenesis of IC/BPS, we studied the changes in cell proliferation, migration, and chemotactic ability of the urothelial cells after LPS-induced inflammatory injury. Our results demonstrated that IL17RA overexpression in urothelial cells suppressed cell proliferation, but did not affect urothelial cell migratory ability after LPS injury (Figure 11). A decrease in cell proliferation due to IL17RA overexpression was observed, which suggests that the bladder epithelial cells’ ability to repair damage may be impaired, which contributes to the denudation of urinary epithelium in IC/BPS. Previous studies using IC/BPS animal models have shown significant upregulation of NLRP3 inflammasome [53,54] and that TLR4/NLRP3 signaling mediates neuroinflammation in IC/BPS [55]. NLRP3 triggers IL1B maturation and secretion. IL1B is a urinary marker and an immunotherapeutic target for IC/BPS [56]. Interestingly, TLR4, NLRP3, and IL1B had similar patterns to IL17RA in consensus clustering. Hence, we explored the association between IL17RA and the TLR4-NLRP3-IL1B pathway. As expected, LPS-induced TLR4-NLRP3-IL1B inflammatory pathway was significantly upregulated after overexpression of IL17RA (Figure 12a), suggesting the increased inflammatory response in urothelial cells under inflammatory stimuli was associated with IL17RA upregulation. Meanwhile, the increased expression of anti-inflammatory factor IL10 and decreased expression of the non-canonical pyroptosis pathway factor CASP4 suggest that IL17RA also initiates a self-protective mechanism in urothelial cells. Finally, we demonstrated that urothelial cells overexpressing IL17RA, stimulated with LPS, triggered phagocytosis, and M2 polarization of macrophages (Figure 12c), which promoted pathological angiogenesis and organ fibrosis [57]. However, additional studies are needed to understand the involvement of IL17RA in volume reduction or glomerulation in the bladder of IC/BPS. In addition, preliminary screening has shown that other factors, GLI1 and BMP4, may mediate the glomerulations and M2 polarization of macrophages. Further, IL17RA overexpression in urothelial cells increased the expression of both the genes (GLI1 and BMP4) after LPS treatment. Previous studies have shown GLI1 promotes vascular remodeling during regeneration after bone injury [58], whereas BMP4 mediates M2 polarization of macrophages in the bladder cancer tumor microenvironment [59]. Taken together, IL17RA may play a dual role in IC/BPS pathogenesis (Figure 12e). On the one hand, it could damage the urothelium by promoting the inflammatory response, on the other hand, it could enhance the anti-inflammatory response by up-regulating anti-inflammatory factors and promoting M2 polarization of macrophages. Hence, IL17RA could be responsible for maintaining a chronic state of inflammation in IC/BPS patients. In sum, the higher expression of IL17RA in IC/BPS patients, in combination with the dual role of IL17RA on IIME, suggested IL17RA as a novel and promising intervention target for IC/BPS patients.

However, there are inevitably some limitations to our study. First, the sample size of the current study is small, due to the lack of publicly available datasets on IC/BPS. Second, some of the normal group tissue samples in this study were from patients diagnosed by the investigators as having diseases unrelated to IC/BPS and may not be representative of completely normal bladder tissue. Thirdly, the molecular mechanism by which IL17RA regulates the TLR4-NLRP3-IL1B pathway and macrophage polarization has not been elucidated in detail. Finally, only overexpression of IL17RA was employed to demonstrate the potential role of this target gene in the pathogenesis of IC/BPS in the current study, although the knockdown of IL17RA would offer additional assistance to further determine its exact function. Over the past 30 years, with the gradual deepening of scientists’ understanding of the mechanism of IL17 family, the efficacy of targeted IL17 family drugs in autoimmune diseases, including psoriasis and ankylosing spondylitis, has been confirmed by more and more studies [50,60]. A recent phase III clinical trial confirmed that the drug brodalumab, which targets IL17RA, has a good therapeutic effect on psoriasis arthritis [61]. Our findings suggest that IL17RA upregulation may be responsible for the long-term chronic inflammation seen in IC/BPS, which encourages us to look further into the role of IL17RA in the pathogenesis of IC/BPS and the possibility of IL17RA targeting drugs in the treatment of IC/BPS. Furthermore, given the complexity and heterogeneity of the pathogenesis of IC/BPS, it is obviously important to identify additional potential therapeutic targets. In addition to IL17RA (including IL17A), we believe that IL1B may be a promising potential therapeutic target for IC/BPS in combination with experimental data and previous research results. Our study found that the IL-1B gene was up-regulated in the bladder tissue of IC/BPS patients, and the overexpression of IL-17RA in urothelial cells significantly up-regulated the expression of this gene. As a key inflammatory factor, IL-1 plays a critical role in autoimmune inflammatory diseases [62], and recent studies have confirmed that IL-1 receptor antagonist has successfully relieved the pain and frequent urination symptoms of IC/BPS patients (13/17) [63]. Therefore, it is worthwhile to further explore the mechanism of IL1B in different IC/BPS patients and accurately apply IL1B targeted drugs to appropriate IC/BPS patients.

## 5. Conclusions

In summary, this study identified different IAMPs in IC/BPS bladder tissue involved in the IIME remodeling of IC/BPS. We also identified IL17RA as a promising biomarker for IC/BPS patients. The dual role of IL17RA mediating both pro- and anti-inflammatory effects may contribute to maintaining the chronic inflammatory state of IC/BPS. IL17RA and related pathways could be used as potential therapeutic targets for IC/BPS.

## Figures and Tables

**Figure 1 biomolecules-13-00421-f001:**
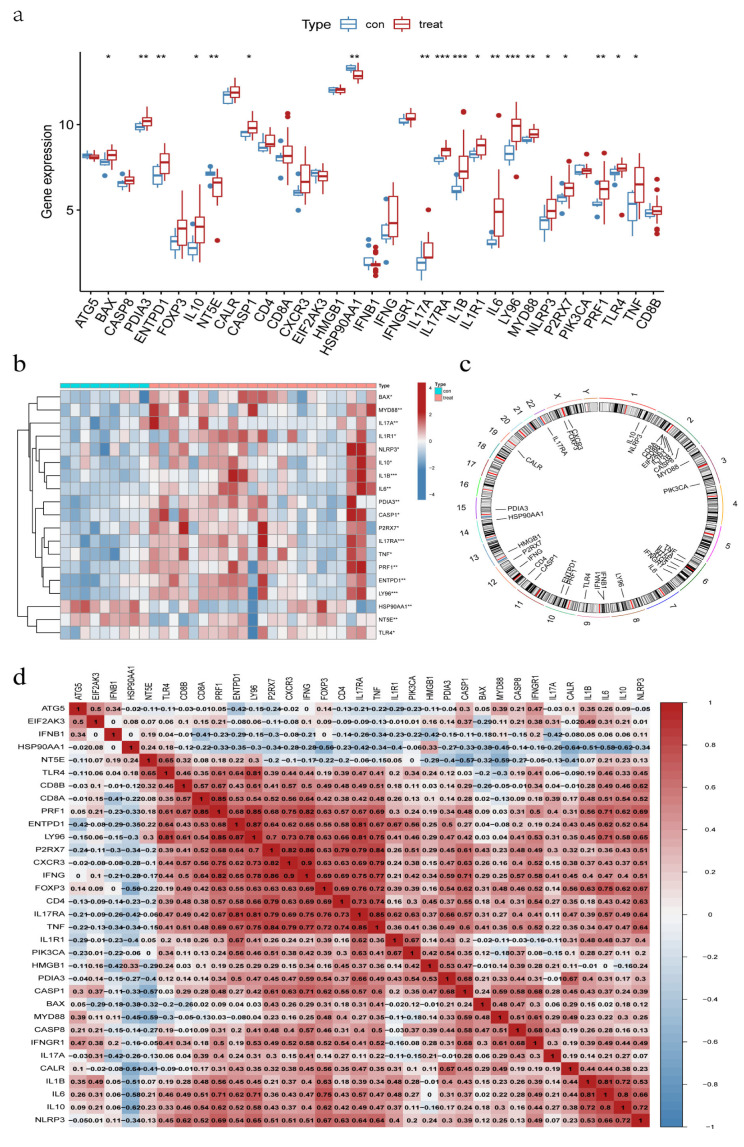
Expression signature of ICD-related genes (IRGs) in IC/BPS bladder tissues. (**a**) Expression of IRGs in IC/BPS tissues; (**b**) Correlation Heatmap of differentially expressed IRGs; (**c**) Location of IRGs on human chromosomes; (**d**) Correlation of IRGs expression in IC/BPS tissues. * Adjust *p* < 0.05, ** Adjust *p* < 0.01, *** Adjust *p* < 0.001.

**Figure 2 biomolecules-13-00421-f002:**
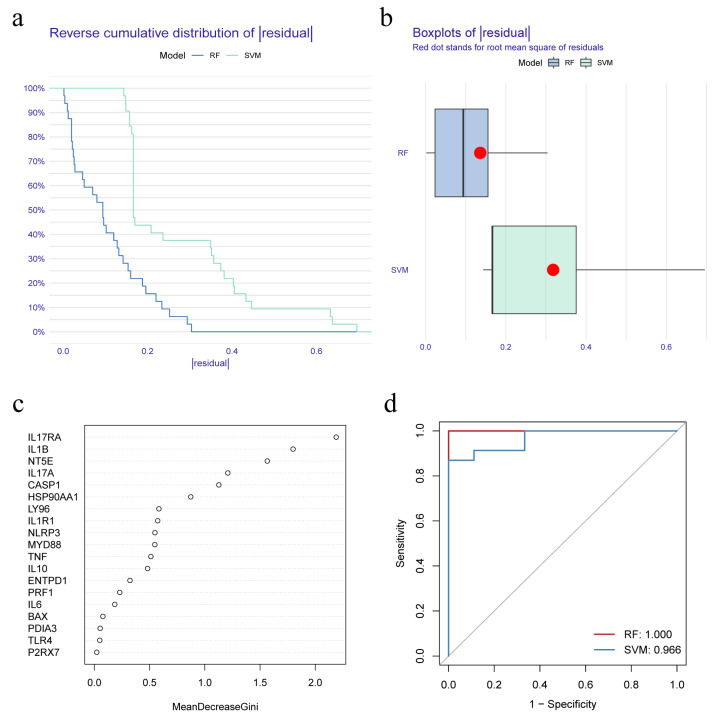
Construction and selection of random forest (RF) and support vector machine (SVM) model. (**a**) Reverse cumulative distribution of residual shows the residual distribution for both RF and SVM models; (**b**) Boxplots of residual shows the distribution of residuals for the RF and SVM models; (**c**) Importance ranking of IRGs based on RF model; (**d**) Receiver operating characteristic (ROC) curves indicate the accuracy of RF and SVM models.

**Figure 3 biomolecules-13-00421-f003:**
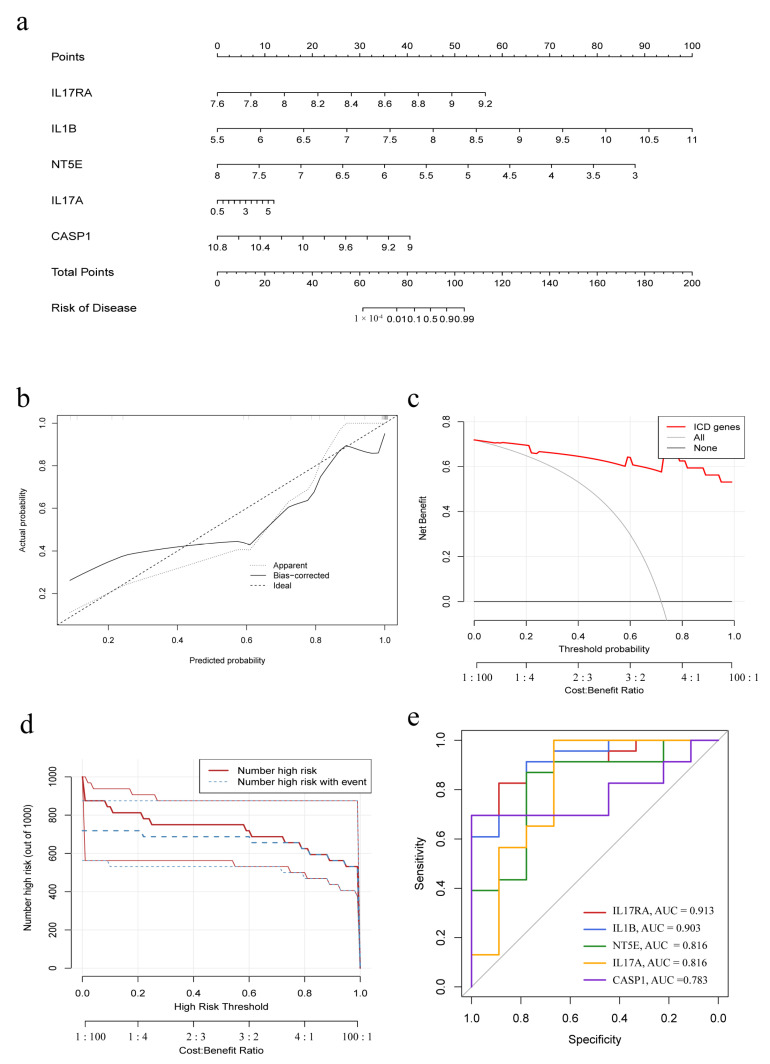
Construction of the nomogram model. (**a**) Nomogram model based on five IRGs; (**b**) Calibration curve embodies the predictive ability of the nomogram model. (**c**) Decision curve analysis indicating that decision-making based on the nomogram model may benefit IC/BPS patients; (**d**) Clinical impact curve to evaluate the predictive ability of the nomogram model; (**e**) Individual predictive power of the five IRGs indicated by ROC curves.

**Figure 4 biomolecules-13-00421-f004:**
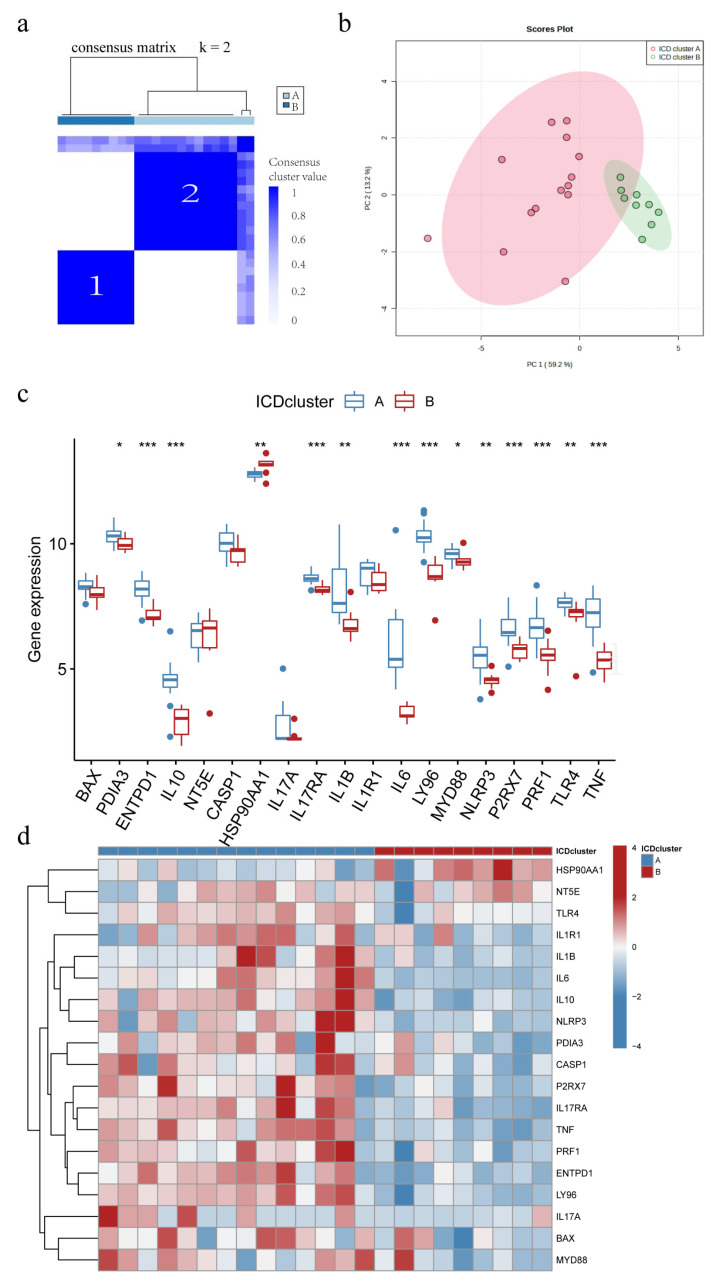
Different ICD-associated molecular patterns (IAMPs) in IC/BPS. (**a**) Consensus clustering of 19 IRGs at k = 2; (**b**) Principal component analysis (PCA) shows ICD cluster A and ICD cluster B; (**c**) Expression boxplots of the 19 significant IRGs in base ICD cluster A and ICD cluster B; (**d**) Expression heatmaps of the 19 significant IRGs in ICD cluster a and ICD cluster B. * *p* < 0.05, ** *p* < 0.01, *** *p* < 0.001.

**Figure 5 biomolecules-13-00421-f005:**
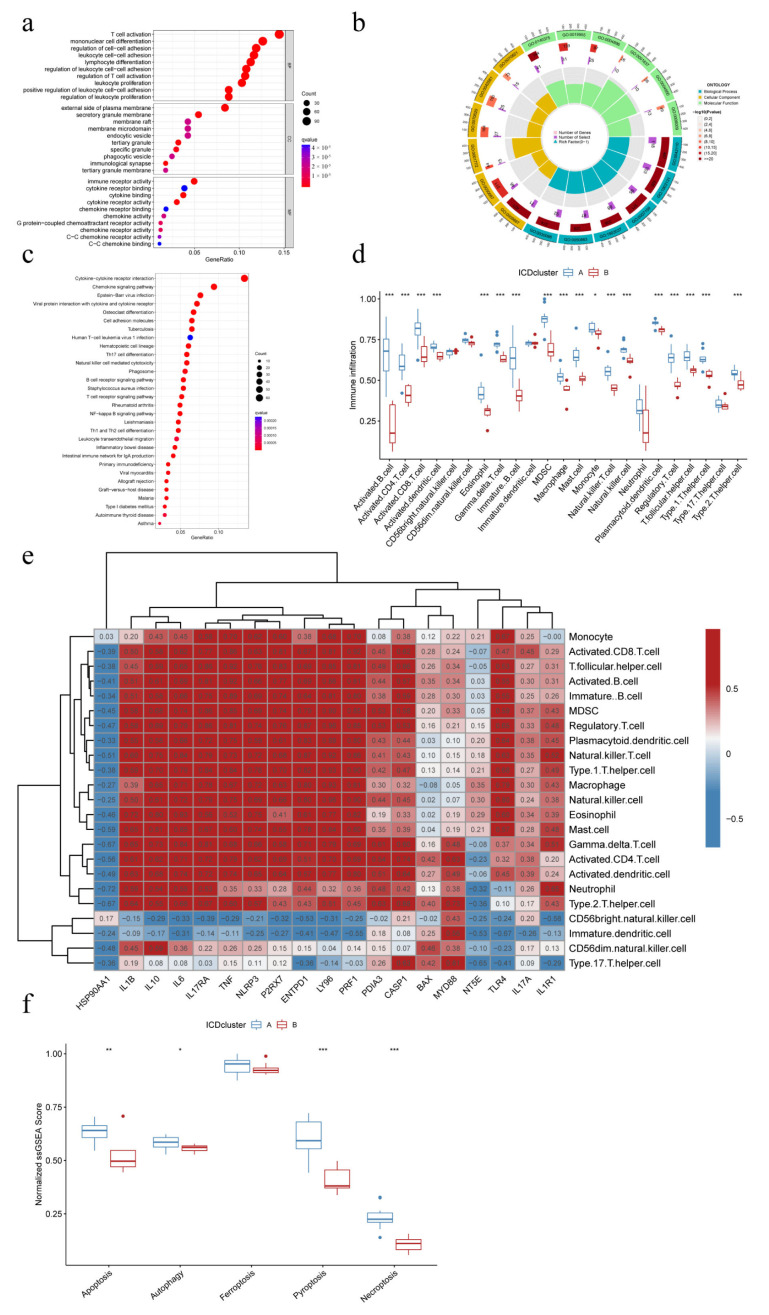
Mechanistic analysis of different IAMPs. (**a**) Gene Ontology (GO) enrichment analysis bubble plot of differential gene expression (DEGs) of different IAMPs; (**b**) GO analysis circle plot of DEGs of different IAMPs; (**c**) Kyoto Encyclopedia of Genes and Genomes (KEGG) pathway enrichment analysis bubble plot of DEGs of different IAMPs; (**d**) Boxplots of the extent of immune cell infiltration in ICD cluster A and ICD cluster B; (**e**) Correlation Heatmap of IRGs and immune cells; (**f**) Expression signature of five common RCD patterns in different IAMPs. * *p* < 0.05, ** *p* < 0.01, *** *p* < 0.001.

**Figure 6 biomolecules-13-00421-f006:**
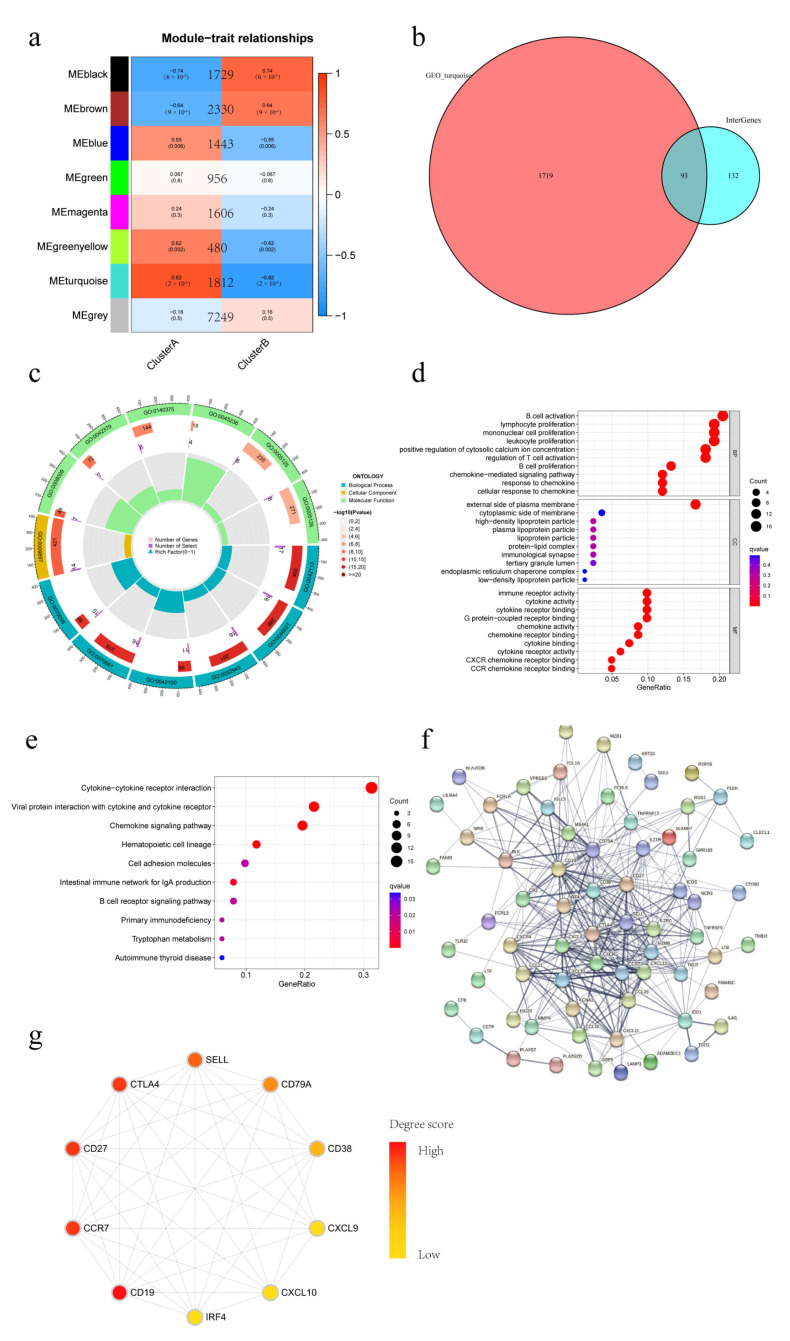
Weighted gene correlation network analysis (WGCNA) versus DEGs Conjoint analysis of different IAMPs. (**a**) WGCNA analysis of different IAMPs; (**b**) Venn plot of intersection genes obtained between WGCNA and DEGs; (**c**) GO analysis circle plot of intersection genes; (**d**) GO analysis bubble plot of intersection genes; (**e**) KEGG analysis bubble plot of intersection genes; (**f**) Protein–protein interaction (PPI) network plot; (**g**) Top ten hub gene network map.

**Figure 7 biomolecules-13-00421-f007:**
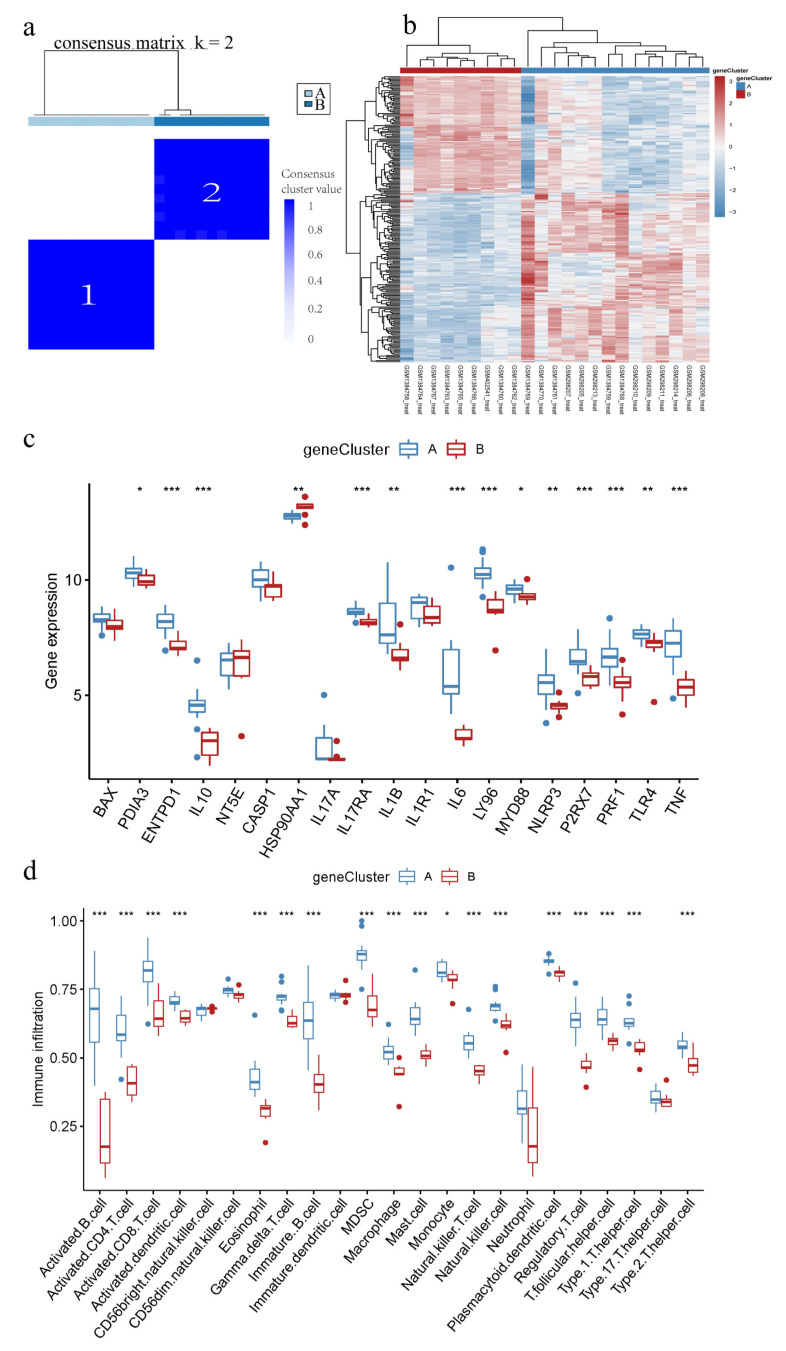
ICD-associated genomic subtypes of IC/BPS. (**a**) Consensus clustering of ICD-associated DEGs at k = 2; (**b**) Heatmap of DEG across the two genomic subtypes; (**c**) Expression boxplots of IRGs in different genomic subtypes; (**d**) Boxplot of immune cell infiltration levels in different genomic subtypes. * *p* < 0.05, ** *p* < 0.01, *** *p* < 0.001.

**Figure 8 biomolecules-13-00421-f008:**
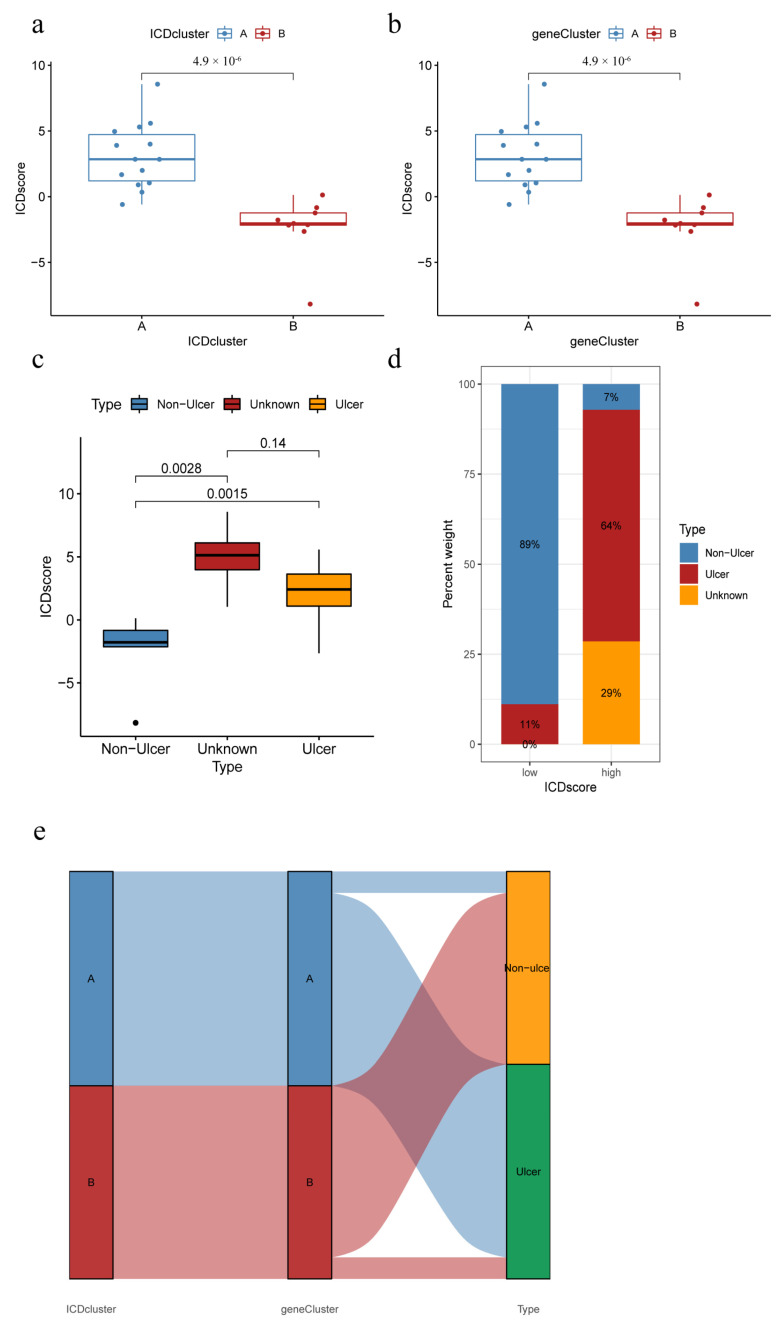
ICD score correlates with clinical subtypes. (**a**) ICD scores of different IAMPs; (**b**) ICD scores for different genomic subtypes; (**c**) ICD score for different IC/BPS clinical subtypes; (**d**) Distribution of IC/BPS clinical subtypes by different ICD score groups; (**e**) Sankey diagram of the relationship between IRGs, IRGs patterns, and ICD scores.

**Figure 9 biomolecules-13-00421-f009:**
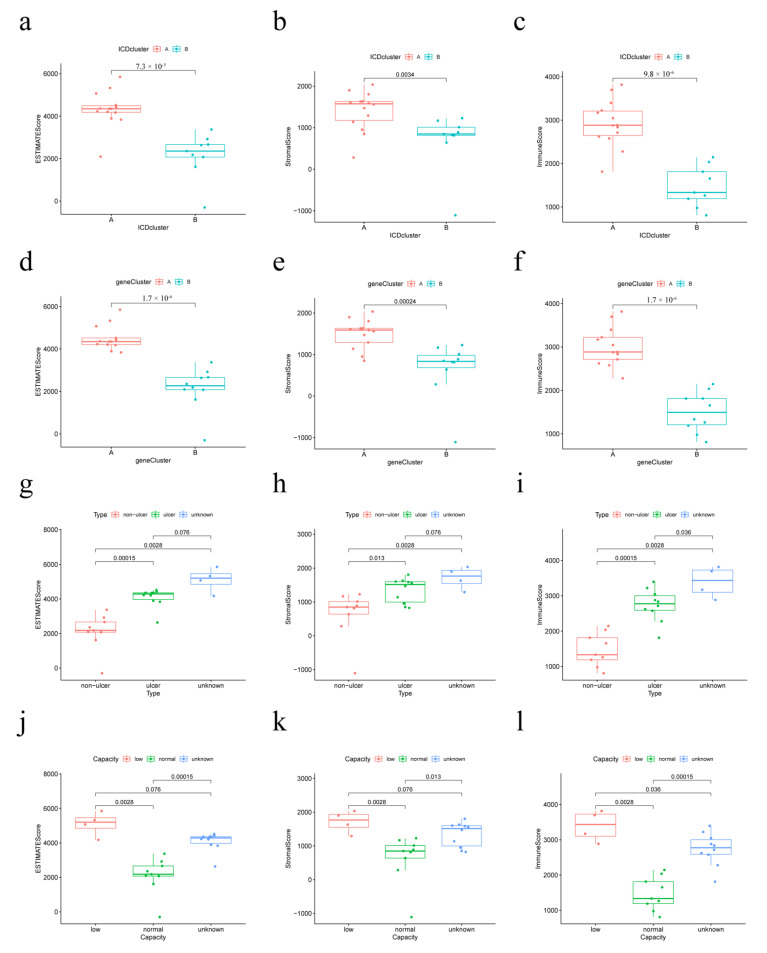
Relationship between immune infiltration score, IAMPs, and clinical features. (**a**–**c**) Immune infiltration scores of different IAMPs; (**d**–**f**) Immune infiltration scores across different genomic subtypes; (**g**–**i**) Immune infiltration scores of different IC/BPS clinical subtypes; (**j**–**l**) Immune infiltration scores in different bladder volume groups.

**Figure 10 biomolecules-13-00421-f010:**
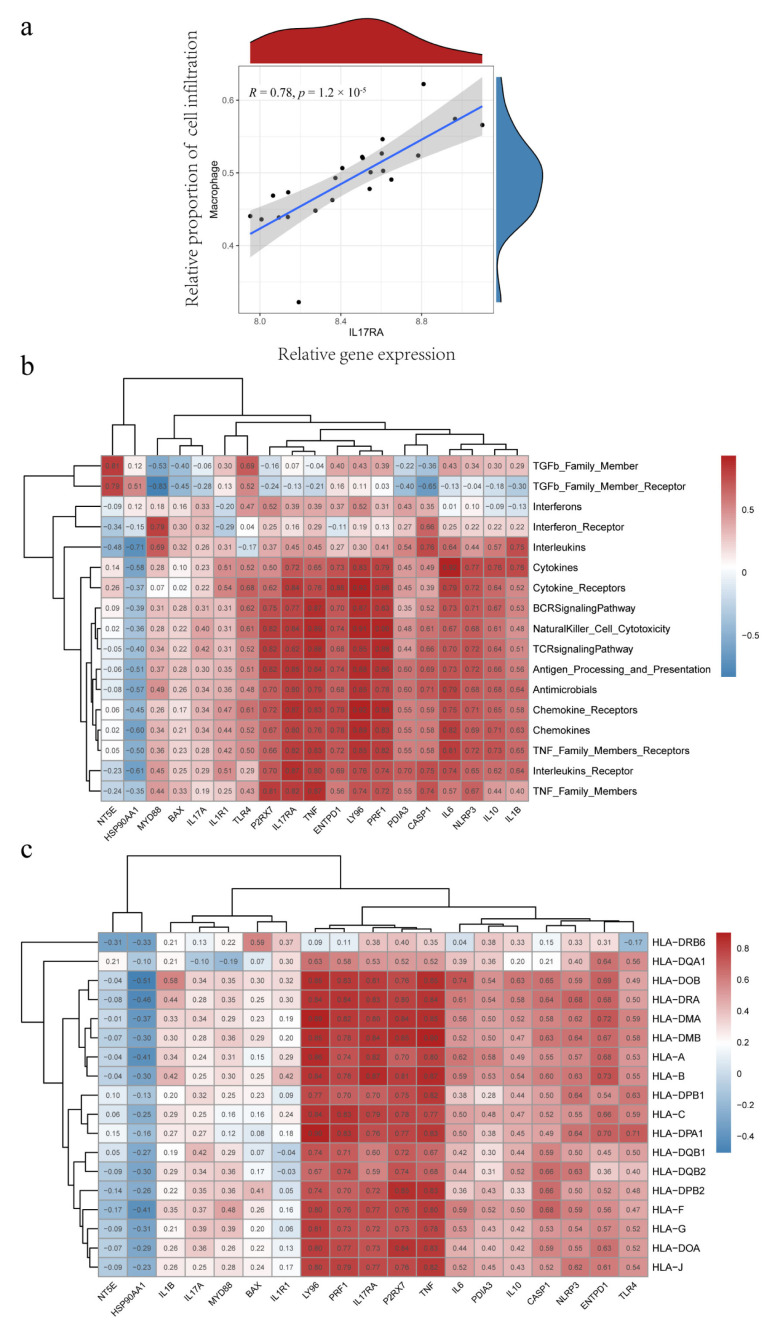
IL17RA correlates with macrophage infiltration, immune function, and HLA gene expression. (**a**) Correlation analysis between IL17RA and macrophage infiltration; (**b**) Correlation analysis between IRGs and immune function; (**c**) Correlation analysis between IRGs and HLA gene expression.

**Figure 11 biomolecules-13-00421-f011:**
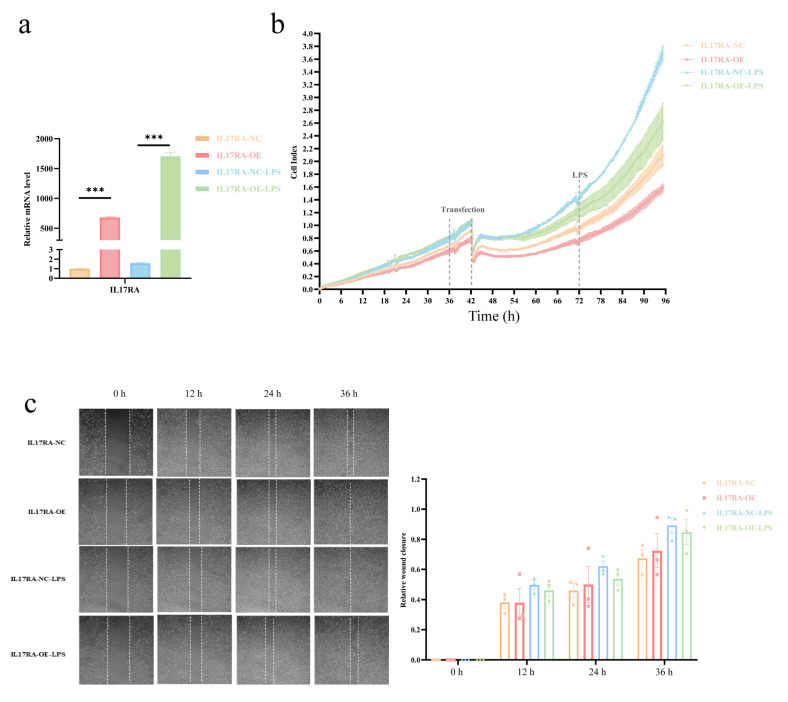
Effect of IL17RA on LPS induced inflammation on proliferation and migration ability of urothelial cells. (**a**) Overexpression efficiency of IL17RA; (**b**) Proliferation curve of urothelial cells; (**c**) Migration ability of urothelial cells was detected by scratch assay. *** *p* < 0.001.

**Figure 12 biomolecules-13-00421-f012:**
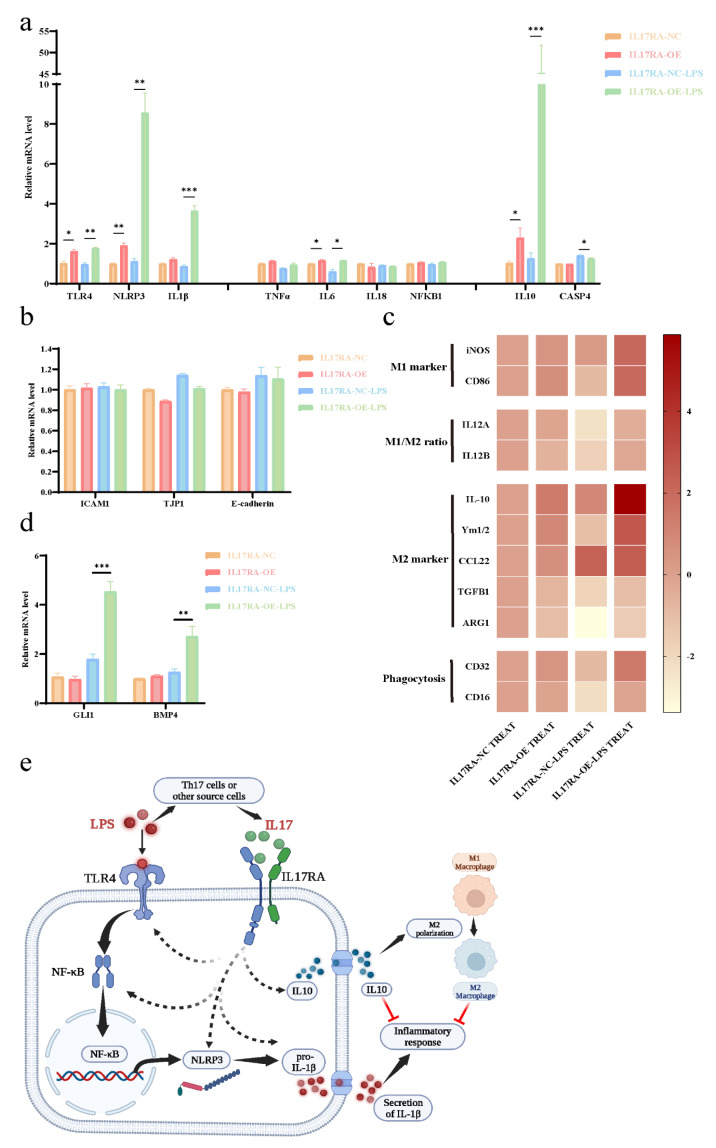
IL17RA regulates the damage repair function of urothelial cells by a dual role. (**a**) Expression of pro- and anti-inflammatory pathway genes; (**b**) Expression of barrier function-related genes; (**c**) Expression heat map (Log2 conversion) of macrophage polarization, phagocytosis genes; (**d**) Expression of GLI1 and BMP4; (**e**) Schematic of the mechanism of IL17RA regulation. * *p* < 0.05, ** *p* < 0.01, *** *p* < 0.001.

## Data Availability

All data used in the study were obtained from the GEO database (https://www.ncbi.nlm.nih.gov/geo/query/acc.cgi?acc=GSE11783,GSE57560, accessed on 1 August 2022).

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
