# Peer review of "Immunogenic Cell Death Associated Molecular Patterns and the Dual Role of IL17RA in Interstitial Cystitis/Bladder Pain Syndrome"

_biomolecules, 2023, doi:10.3390/biom13030421_

Round 1
Reviewer 1 Report
Well written with minor difficulties on native English writing
More samples/patients, should be considered in the future diferences between races and continents. Only samples from 23 patients, extensively studied
Potential therapeutic use of IL17RA should be better fundamental in the conclusion
Conclusions about potential targets should be more emphasised. Also what really could be potential applications in clinical practice
Author Response
Reviewer 1:
Well written with minor difficulties on native English writing
Reply:Thank you for your review and valuable comments on our research.
More samples/patients, should be considered in the future differences between races and continents. Only samples from 23 patients, extensively studied
Reply:Thank you for your reminding and suggestions. The sample size of this study is small, and there is a lack of exploration of the differences between races and continents. These two problems are indeed the directions we need to improve in the future. In the next further study, we will enroll more patients and collect more diversified samples (including bladder tissue, blood, urine, etc.), hoping to get more promising research results.
Potential therapeutic use of IL17RA should be better fundamental in the conclusion. Conclusions about potential targets should be more emphasised. Also what really could be potential applications in clinical practice.
Reply:Thank you for your suggestion and we strongly agree with you. We have added the discussion about the potential targets and the clinical application of IL17RA (Line:613-632:Over the past 30 years, with the gradual deepening of scientists' understanding of the mechanism of IL17 family, the efficacy of targeted IL17 family drugs in autoimmune diseases including psoriasis and ankylosing spondylitis has been confirmed by more and more studies [50,60]. A recent phase III clinical trial confirmed that the drug brodalumab, which targets IL17RA, has a good therapeutic effect on psoriasis arthritis [61]. Our findings suggest that IL17RA upregulation may be responsible for the long-term chronic in-flammation seen in IC/BPS, which encourages us to look further into the role of IL17RA in the pathogenesis of IC/BPS and the possibility of IL17RA targeting drugs in the treatment of IC/BPS. Furthermore, given the complexity and heterogeneity of the pathogenesis of IC/BPS, it is obviously important to identify additional potential therapeutic targets. In addition to IL17RA (including IL17A), we believe that IL1B may be a promising potential therapeutic target for IC/BPS in combination with experimental data and previous re-search results. Our study found that IL-1B gene was up-regulated in bladder tissue of IC/BPS patients, and overexpression of IL-17RA in urothelial cells significantly up-regulated the expression of this gene. As a key inflammatory factor, IL-1 plays a critical role in autoimmune inflammatory diseases [62], and recent studies have confirmed that IL-1 receptor antagonist has successfully relieved the pain and frequent urination symptoms of IC/BPS patients (13/17) [63]. Therefore, it is worthwhile to further explore the mechanism of IL1B in different IC/BPS patients and accurately apply IL1B targeted drugs to appropriate IC/BPS patients.). We thank you for your valuable suggestions on our research and look forward to receiving your guidance and review again!

Reviewer 2 Report
Although the authors were not responsible for the GEO datasets, a crucial item is what was selected to represent Normal. The "Normals" in many bladder studies actually are based on "normal appearing" urothelium, whicch may or may not be normal. To the extent the authors can determine the source of "normal" tissue for their studies, this needs a comment.
The manuscript uses numerous acronyms. A list of these and their full meanings would be helpful.
The bioinformatic analysis is state of the art and correlates nicely with clinical and scientific observations.
Author Response
Reviewer 2:
Although the authors were not responsible for the GEO datasets, a crucial item is what was selected to represent Normal. The "Normals" in many bladder studies actually are based on "normal appearing" urothelium, which may or may not be normal. To the extent the authors can determine the source of "normal" tissue for their studies, this needs a comment.
Reply:Thank you for your suggestions on our research. We agree with you very much. Our study included two data sets, GSE57560 and GSE11783. The original articles provided the introduction of the control group.
1、GSE57560:“Control subjects were drawn from the population of female patients who presented to the same urology clinic for urological evaluation requiring biopsy unrelated to IC/BPS”(Colaco M, Koslov DS, Keys T, Evans RJ, Badlani GH, Andersson KE, Walker SJ. Correlation of gene expression with bladder capacity in interstitial cystitis/bladder pain syndrome. J Urol. 2014 Oct;192(4):1123-9. doi: 10.1016/j.juro.2014.05.047. Epub 2014 May 17. PMID: 24840534.)
2、GSE11783:“Four control patients did not have any symptoms related to IC, but underwent vaginal prolapse surgery after biopsy removal. Two control patients, patient 2 and patient 10, had bladder-related symptoms. Patient 2 had urgency and frequency for 20 months, only minor pain, and a bladder capacity of 400 ml. Thus, this patient meets the relaxed eligibility criteria for the IC Database Study”(Gamper M, Viereck V, Geissbühler V, Eberhard J, Binder J, Moll C, Rehrauer H, Moser R. Gene expression profile of bladder tissue of patients with ulcerative interstitial cystitis. BMC Genomics. 2009 Apr 28;10:199. doi: 10.1186/1471-2164-10-199. PMID: 19400928; PMCID: PMC2686735.)
Therefore, the "normal" issue included in this study does have certain limitations. According to your suggestion, we added the corresponding content in the discussion section(Line 606-608: some of the normal group tissue samples in this study were from patients diagnosed by the investigators as having diseases unrelated to IC / BPS and may not be representative of completely normal bladder tissue).
The manuscript uses numerous acronyms. A list of these and their full meanings would be helpful.
Reply:Thank you for your reminder. We have added the file “Abreviations” to the supplementary material.
The bioinformatic analysis is state of the art and correlates nicely with clinical and scientific observations.
Reply:We appreciate your recognition of our research and look forward to your review and guidance again!

Reviewer 3 Report
The authors describe here an interesting analysis of Immunogenic Cell Death Associated Molecular Patterns (IAMPs) in Interstitial Cystitis/Bladder Pain Syndrome (IC/BPS). Following their analysis the authors identified IL17RA as a promising biomarker for IC/BPS patients. Although the authors have made a thorough analysis of existing database and combined them with original results, several important issues must be addressed before considering the manuscript for publication.
Major:
1. The authors should add in the manuscript description regarding the selection of their analytical methodologies and the biological motivation for performing all this kind of analyses. Moreover the authors also frequently confound the plots as they report histogram but actually showing boxplots (see for exp. Figures 4c, 7c,d, etc.).
2. The authors report DE analysis of IRGs between normal and IC/BPS samples using the Wilcoxon test, taking p<0.05 as significance threshold. This is a robust test, based on ranking the data. However, they analyze multiple genes in the same dataset so they should disclose the p-value correction used for assessing the significant differences.
3. Figure 1b: The heat map displays values between -4 and +4. What kind of correlation do the authors use for this figure?
4. PCA analysis in Figure 4b suggests greater variability in cluster A compared to cluster B, particular for the second dimension, PC2, where clearly there is at least a subgroup of patients with lower values compared with the rest of patients. This is also suggested by judging by the displayed axis scale. I would suggest the authors to at least provide some comments regarding this aspect and change the axis in %, so that the reader can better judge the contribution of each dimensionality to the total variance.
5. GO analysis in Figures 5a-c displays counts of 30, 60, 90 etc. How can this be since the authors report GO analysis of DEGs in A and B cluster, counting a total of 19 genes. The authors should either explain in detail how they performed this analysis (particular what was considered as an input) or motivate their rationale for performing such an analysis or they should remove it, since it does not bring any value to the manuscript. It is clear that the displayed DEGs in clusters A and B are related to immune response since most of the genes are cytokines or receptors for these.
6. How did the authors assess immune infiltration? Which parameters were used? This should be described in detail.
7.Overall figure 12 is confusing, it is very unlikely that there is a very small difference between non-treated and LPS-treated cells in terms of TLR4, NLRP3 and IL1B mRNA expression level in the first place. Please indicate more details in the figure legends, for example for fig 12C. Also, in figure 12E, the mechanistic details presented are not totally supported by the data presented in the manuscript.
8.For the experiments assessing the interactions between urothelial cells and macrophages, the combination of human with mouse cell lines is not desirable since not all proteins have the same structure to be able to bind the receptors on the receiver cells and secondly Raw264.7 cells lack the caspase-1 gene, therefore the NLRP3-IL-1B axis is impaired in these cells, so the results obtained should be corroborated with other using human macropahges to avoid any false positive/negative results
9. Please add more details regarding the experimental design throughout the entire manuscript, it will add value to the article.
Minor:
1. Please provide a relevant graph, which shows that the merged dataset is not inflicted by the batch effect.
2. Figure 1d: There are some clusters observed in this figure. For example there are clusters TLP4-TNF, IL1B-NLRP3, etc. Do the authors use clustering when displaying the data? If yes this should be disclosed. Also please re-upload a higher resolution figure, as the gene names are barely visible.
3. How many genes were in each WGCNA module? Please report in the corresponding figure.
4. Figure 6f-g: What does line thickness denotes? How about the colors in Figure 6g? Please add the corresponding legends.
5. Figure 7a: Please add legend to the figure? What does the blue color denotes? Why k=2 when there are three clusters in the figure?
6. Figure 7b: It looks like the first sample in each cluster is skewed compared with the others. Why? This is also evident in the clusters.
7. Figure 10a: Please add units for both axes of the figure.
Author Response
Reviewer 3:
The authors describe here an interesting analysis of Immunogenic Cell Death Associated Molecular Patterns (IAMPs) in Interstitial Cystitis/Bladder Pain Syndrome (IC/BPS). Following their analysis the authors identified IL17RA as a promising biomarker for IC/BPS patients. Although the authors have made a thorough analysis of existing database and combined them with original results, several important issues must be addressed before considering the manuscript for publication.
Reply:Dear reviewer, we are very grateful for your valuable comments. We updated the manuscript according to your suggestions. We hope that we have correctly understood your questions. If there is any misunderstanding or incorrect modification, we look forward to receiving your further guidance.
Major:
- The authors should add in the manuscript description regarding the selection of their analytical methodologies and the biological motivation for performing all this kind of analyses. Moreover the authors also frequently confound the plots as they report histogram but actually showing boxplots (see for exp. Figures 4c, 7c,d, etc.).
Reply:Thank you for your reminder. We are very sorry for confusing the names of the chats. We have corrected the error of the figure legends.
Our study does not recount the biological motivation for the plethora of bioinformatics analyses for reasons such as the following.
Our study conducted an initial bioinformatics analysis based on the geo public database, and the bioinformatics analysis method used in this study was widely adopted by many related studies with already published geo databases(e.g., RF, SVM, ssGSEA, GSVA, etc.). Elaboration of the biological motivation for each bioinformatics analysis approach has rarely been seen in these numerous studies. For example, two other studies in this section also did not address related issues. So, we would like to ask for your consent not to supplement the above information. Please forgive us~
(1. Lu, Q.; Nie, R.; Luo, J.; Wang, X.; You, L. Identifying Immune-Specific Subtypes of Adrenocortical Carcinoma Based on Immunogenomic Profiling. Biomolecules 2023, 13, 104.
- Yang, Y.; Luo, D.; Gao, W.; Wang, Q.; Yao, W.; Xue, D.; Ma, B. Combination Analysis of Ferroptosis and Immune Status Predicts Patients Survival in Breast Invasive Ductal Carcinoma. Biomolecules 2023, 13, 147.)
- The authors report DE analysis of IRGs between normal and IC/BPS samples using the Wilcoxon test, taking p<0.05 as significance threshold. This is a robust test, based on ranking the data. However, they analyze multiple genes in the same dataset
Reply:Your recommendation is very correct. We have made an incorrect presentation here. We have made corrections. (Line 76: To correct the p value, Benjamini and Hochberg (BH) adjustment was used. Adjust P < 0.05 was set as the threshold for selection.)
- Figure 1b: The heat map displays values between -4 and +4. What kind of correlation do the authors use for this figure?
Reply:Thank you for your question. We normalized the expression data of genes by row (same gene), so values ranged between - 4 and + 4.
- PCA analysis in Figure 4b suggests greater variability in cluster A compared to cluster B, particular for the second dimension, PC2, where clearly there is at least a subgroup of patients with lower values compared with the rest of patients. This is also suggested by judging by the displayed axis scale. I would suggest the authors to at least provide some comments regarding this aspect and change the axis in %, so that the reader can better judge the contribution of each dimensionality to the total variance.
Reply:Thank you for your suggestions. We have updated the Figure as you suggested (see new Figure 4b) and added the corresponding content in the discussion session. (Line524-527:It was also noteworthy that PCA analysis showed that ICD cluster A had greater in-tragroup variability. Combined with the association between ICD clueter A and HIC subtypes, we have reason to speculate that different subgroups may still exist within HIC patients.)
- GO analysis in Figures 5a-c displays counts of 30, 60, 90 etc. How can this be since the authors report GO analysis of DEGs in A and B cluster, counting a total of 19 genes. The authors should either explain in detail how they performed this analysis (particular what was considered as an input) or motivate their rationale for performing such an analysis or they should remove it, since it does not bring any value to the manuscript. It is clear that the displayed DEGs in clusters A and B are related to immune response since most of the genes are cytokines or receptors for these.
Reply:We are very sorry that this part of the content makes you confused. First of all, it should be noted that the analysis in Figure 5a-c is not based on 19 differentially expressed IRGs. We are dividing all IC/BPS patients into ICD cluster A and B through the expression of differential IRGs (see Figure 4). Next, we reanalyzed all differentially expressed genes (not only IRGs) between ICD cluster A and B patients. Then all 914 differentially expressed genes were analyzed by GO and KEGG. That is to say, the input data of the analysis in Figure 5a-c are all differentially expressed genes (914 in total) between patients in ICD cluster A and B groups. We hope our explanation answers your question. We apologize again.
- How did the authors assess immune infiltration? Which parameters were used? This should be described in detail.
Reply:We appreciate your suggestion very much. We have added a brief description of the evaluation method of immune infiltration. (Line131-134: In short, we used the Estimate algorithm to evaluate the different components of bladder tissue in patients with IC/BPS. The stromal score represents the matrix component of bladder tissue, the immune score represents the infiltration of immune cells in bladder tissue, and the Estimate score represents the purity of bladder tissue)
7.Overall figure 12 is confusing, it is very unlikely that there is a very small difference between non-treated and LPS-treated cells in terms of TLR4, NLRP3 and IL1B mRNA expression level in the first place. Please indicate more details in the figure legends, for example for fig 12C. Also, in figure 12E, the mechanistic details presented are not totally supported by the data presented in the manuscript.
Reply:We are very sorry for not showing this part clearly.
First, in response to your question "It is very unlikely that there is a very small difference between non-treated and LPS-treated cells in terms of TLR4, NLRP3 and IL1B mRNA expression level in the first place." Our explanation is as follows.
As we described in section 2.13 (Interactions between urothelial cells and macrophages), we treated SV-HUC-1 cells with LPS for 24 hours, and then gave them the medium without LPS for another 12 hours. SV-HUC-1 cells were detected by QPCR, and macrophages were treated with the obtained conditioned medium. Because there is no evidence of long-term persistent bacterial infection in IC/BPS patients, we used the above operation to simulate the self-recovery process after transient injury.
Although we have not found a study that is completely consistent with our cell processing method, we have noticed that several published articles can support our results from different perspectives.
- The research of Janicova et al. confirmed that the concentrations of TLR4 and IL-1B secreted by blood cells treated with LPS were lower than the baseline at 24h and 48h, both in vivo and in vitro.
(Janicova A, Haag F, Xu B, Garza AP, Dunay IR, Neunaber C, Nowak AJ, Cavalli P, Marzi I, Sturm R, Relja B. Acute Alcohol Intoxication Modulates Monocyte Subsets and Their Functions in a Time-Dependent Manner in Healthy Volunteers. Front Immunol. 2021 May 18;12:652488. doi: 10.3389/fimmu.2021.652488. PMID: 34084163; PMCID: PMC8167072.)
- Kent et al's research found that after LPS treatment of primary Bovine Ruminal Epithelial cell, the culture medium was replaced with one without LPS. TLR4 mRNA expression returned to baseline level.
(Kent-Dennis C, Aschenbach JR, Griebel PJ, Penner GB. Effects of lipopolysaccharide exposure in primary bovine ruminal epithelial cells. J Dairy Sci. 2020 Oct;103(10):9587-9603. doi: 10.3168/jds.2020-18652. Epub 2020 Jul 31. PMID: 32747102.)
Therefore, it is acceptable that the expression level of TLR4, NLRP3 and IL1B mRNA does not increase from the baseline after 24 hours of LPS treatment and 12 hours of replacement of fresh culture medium.
I hope our explanation can answer your question.
In addition, we changed the incompletely confirmed part of Fig12E into a dotted line and prepared to further confirm it in the next study.
8.For the experiments assessing the interactions between urothelial cells and macrophages, the combination of human with mouse cell lines is not desirable since not all proteins have the same structure to be able to bind the receptors on the receiver cells and secondly Raw264.7 cells lack the caspase-1 gene, therefore the NLRP3-IL-1B axis is impaired in these cells, so the results obtained should be corroborated with other using human macropahges to avoid any false positive/negative results
Reply:Thank you for your precious suggestions, and we agreed that it should be better to use human macrophages (like THP-1) for research, which could improve the reliability of the result. So, we purchased the THP-1 in the first place and performed the conditioned media-co-culture assay using the human macrophages. Unfortunately, we encountered a Gordian knot concerning cultivating the cell. Even though we have repeatedly adjusted the medium serum ratio (Gibco, 10-20%) and cell density, the cells were still in poor condition, grew slowly, and couldn’t be induced into macrophage phenotype by PMA. Part of the results was shown below, and we would appreciate your suggestions to help us solve the problem. We have ordered a new cell line for more trying. We’re so sorry for not completing this part of the experiment currently. Once the condition permits, we would love to retry this part of the experiment and put the results in our following in-depth research regarding IL17RA, about which we hope to have the chance to communicate with you. Besides, the interaction between SV-HUC-1 and RAW264.7 cells could still demonstrate that IL17RA could influence the polarization of macrophages to some extent. Since this current research was only the preliminary exploration of the relationship between ICD and IC/BPS, we are looking forward to communicating with you both on the methodology of media-co-culture assay and on our following in-depth research on IL17RA. Thanks again for your valuable suggestion!
- Please add more details regarding the experimental design throughout the entire manuscript, it will add value to the article.
Reply:Thank you for your suggestion. We have added some experimental design content.(Line189-197:2.14. Chronic LPS stimulation's influence on urothelial cells
As previously described:after 36 h of transfection, four groups of SV-HUC-1 cells were treated with vehicle(IL17RA-NC group, and IL17RA-OE group) or LPS(5μg/ml,IL17RA-NC-LPS group, and IL17RA-OE-LPS group), respectively. The medium was changed to fresh medium without LPS after 24 h, and the incubation was continued for another 12 h. The purpose of the above treatments was to mimic the microenvironment in which cells recovered from LPS injury. Finally, the changes of pro-inflammatory, anti-inflammatory and barrier function related genes in urothelial cells of different groups were examined.)
Minor:
- Please provide a relevant graph, which shows that the merged dataset is not inflicted by the batch effect.
Reply:Thank you for your suggestion, we used the "combat" function in the SVA package to remove the batch effect.
- Figure 1d: There are some clusters observed in this figure. For example there are clusters TLP4-TNF, IL1B-NLRP3, etc. Do the authors use clustering when displaying the data? If yes this should be disclosed. Also please re-upload a higher resolution figure, as the gene names are barely visible.
Reply:We do not use clustering in Figure 1d. Meanwhile, Figure 1 has been updated.
- How many genes were in each WGCNA module? Please report in the corresponding figure.
Reply:Thank you for your reminder. Figure 6a has been updated
- Figure 6f-g: What does line thickness denotes? How about the colors in Figure 6g? Please add the corresponding legends.
Reply:We are very sorry. There is no difference in the thickness of the lines in Figure 6f. The color in Figure 6g from red to yellow represents the degree score of the hub gene. The corresponding legend has been added.
- Figure 7a: Please add legend to the figure? What does the blue color denotes? Why k=2 when there are three clusters in the figure?
Reply:Thank you for your reminder. The legend has been added. The blue color represents the Consumer Clustering value. In addition, when K=2, there are two clusters, which we have marked in the figure.
- Figure 7b: It looks like the first sample in each cluster is skewed compared with the others. Why? This is also evident in the clusters.
Reply:Thanks for your guidance, we observed this phenomenon. We conjecture that grouping by gene cluster has not yet achieved perfect discrimination, which is also evident in the correspondence in the Sankey diagram (fig8e)
- Figure 10a: Please add units for both axes of the figure.
Reply:Thanks for your suggestion, we have updated Fig10a。

Round 2
Reviewer 3 Report
The authors have addressed all my concerns, in this format I recommend the publication of the manuscript.